# In-Depth Chemical Characterization of *Punica granatum* L. Seed Oil

**DOI:** 10.3390/foods12081592

**Published:** 2023-04-09

**Authors:** Francesco Cairone, Chiara Salvitti, Antonia Iazzetti, Giancarlo Fabrizi, Anna Troiani, Federico Pepi, Stefania Cesa

**Affiliations:** 1Department of Drug Chemistry and Technology, “Sapienza” University of Rome, 00185 Rome, Italy; 2Department of Basic Biotechnological Sciences, Intensive and Perioperative Clinics, Catholic University of the Sacred Heart, 00168 Rome, Italy; 3Policlinico Universitario ‘A. Gemelli’ Foundation-IRCCS, 00168 Rome, Italy

**Keywords:** pomegranate seed oil, HPLC-DAD analysis, AP-MALDI-MS analysis, ^1^H and ^13^C-NMR analysis

## Abstract

Fruit seeds belonging to the pomegranate cultivar “Granata” were subjected to extraction and oily component analysis, with the aim of obtaining information about their composition. The presence of conjugated isomers of linolenic acid (CLNA isomers) in the oily phase extracted from the seeds gives a high added value to this part of the fruit, which is too often considered and treated as waste. The separated seeds were subjected to a classic Soxhlet extraction with *n*-hexane or extraction with supercritical CO_2_, assisted by ethanol. The resulting oils were evaluated by ^1^H and ^13^C-NMR and AP-MALDI-MS techniques. Differences in the triacylglycerols composition, with particular regard to punicic acid and other CLNA content, were studied in depth. Results showed the prevalence of punicic acid in the triacylglycerol mixture up to the 75%, with clear preponderance in the extract by supercritical fluids. Consequently, other CLNA isomers are, altogether, two-fold less represented in the supercritical extract than in the Soxhlet one. The two oily residues were subjected to solid phase extraction (SPE) and to HPLC-DAD analysis for the polyphenolic isolation and characterization. In addition to HPLC analysis, which showed different content and composition, DPPH analysis to evaluate the antiradical potential showed that the extract obtained with supercritical CO_2_ was much more active.

## 1. Introduction

Pomegranate (*Punica granatum* L.) is a well-known tree, cultivated since ancient times for its edible fruits and its symbolic meaning. More than 500 different varieties are nowadays diffused and cultivated in all the world regions with Mediterranean climatic conditions. Regardless of the significant differences among cultivars, all the edible parts of fruits, including the peels and other not-edible parts, are considered of high healthy potential and are widely studied [1]. Different recent reviews underline once again the rich chemical composition of pomegranate peels (mainly containing hydrolysable ellagitannins, minerals, and polysaccharides), arils (free sugars, organic acids, phenolics, flavonoids, anthocyanins, and pectins, besides about 85% of water), and seeds (protein, fibres, pectins, polyphenols, and isoflavones, besides about 15% of triglyceride oil), reiterating the exerted health benefits [1]. Antioxidant, anti-inflammatory, anti-mutagenic, cytoprotective, anti-angiogenic, and anticancer activities are appointed to the interesting phytocomplex of pomegranate. Besides the different exerted activities, anti-hypertensive, cardiovascular and neuroprotective, antidiabetic, antibacterial, and antifungal properties were tested in vitro and in vivo and summarized in recent reviews [2,3]. A review [4] also focused on the healthy potential of pomegranate seeds, confirming the healthy anti-inflammatory, anticancer, antimicrobial, and antioxidant potential of this fruit part. Authors also underline the fact that pomegranate seeds, while representing an edible part of pomegranate fruit, are often downgraded to food waste, particularly when juices are prepared as centrifuged products and not by squeezing. Pomegranate fruits or separated peels, coming from different Italian varieties, comprising the Dente di Cavallo (DC) which represents the most diffuse Italian cultivar, were evaluated in some previous papers, for their phytochemical composition and relative healthy potential. Chemico-physical properties, pH and color, total soluble solids content and sugar/acid ratio, vitamin C, punicalagin, ellagic acid, and anthocyanins contents were evaluated. The pomological and physicochemical studies were also performed with the aim of better characterizing the pomegranate biodiversity [5,6,7], showing that samples displayed a high variability. In any case, pomegranate fruits represented an excellent source of minerals, especially potassium, recognized for the key role played in the cells. Moreover, high anti-radical activity was correlated with the total phenolic content, especially characterized by cyanidin- and pelargonidin-3,5-diglucosides as the most represented anthocyanins [8]. The pomegranate wastes, such as marcs coming from squeezed pomegranate or seeds coming from centrifuged juices, represent by-products exceptionally rich in bioactive compounds and excellently reused in formulated products which find application in cosmetic and pharmaceutic yields, as well as in food supplements [9]. A recent review deals with the effect of pomegranate seed oil on different diseases [10] and two others [11,12] are, in particular, focused on the bioactive properties of conjugated isomers of α-linolenic acid (CLNA isomers). These papers report that molecules such as punicic acid, highly contained in pomegranate seed oil, play a role in immune function, with effects on eicosanoid production and suppression of inflammatory response. Effects were also shown in the prevention of cardiovascular disease and in delaying the onset and progression of obesity. They were also found active against polycystic ovary syndrome, an increasing clinical event, correlated with amenorrhea and infertility. Moreover, these bioactive are reported to exhibit a strong cytotoxic effect, preventing cancer and reducing the triacylglycerol accumulation in the liver, and regulating the LDL cholesterol levels in the blood [13].

On the other hand, a rich polyphenolic profile has also been identified. In particular, in a recent review [4] conducted on more than 15 pomegranate seeds’ cultivar, different flavonoids (such as catechin, dihydroxygallocatechin, naringenin hexoside, quercetin-3-rhamnoside, quercetin hexoside, kaempferol-3-glucoside) and ellagic acid, which represents the major hydrolysable tannin in pomegranate seeds were identified.

Anyway, the oil quality is strongly influenced by the adopted extraction procedure.

For our investigation, we focused on two main methods: the classical Soxhlet extraction and an unconventional procedure based on the use of supercritical CO_2_. In the Soxhlet extraction process, non-polar organic solvents are required for the extraction of the lipid phase. In dependence on the solvent nature, temperatures of about 30 °C, for prolonged times (6–8 h, extraction yields of 13–20%), up to temperatures above 90 °C, for very short times (3 min, extraction yields of 35%), were tested [14,15,16,17]. Various disadvantages are commonly linked to this methodology. Long extraction times and/or high temperatures compromise the preservation of bioactive labile compounds. Moreover, the use of expensive and environmentally unfriendly organic solvents is required [18]. These drawbacks may be conveniently overcome by adopting the cold-press extraction and the supercritical fluids methods.

The cold press extraction method has many advantages in terms of time consumption and cost [19]. Moreover, according to the literature data, it appears as the best method in terms of control of acidity and peroxide value, also accompanied by a higher phenolic content and a better aromatic profile [16,17]. However, quite low extraction yields are reported, so other methods could be preferred for industrial applications.

Supercritical CO_2_ extraction if compared to classical lipid extraction methods is recognized as faster, more selective, and more efficient. In addition, this approach may be considered preferable in terms of yield, effectiveness, and quality, other than representing a clean and safe technique for the environment [20,21,22].

For these reasons, considering the healthy potential associated with pomegranate derivatives, seeds of Granata cv were extracted. In the frame of a renewed interest towards the transformation of waste into high-added products, the aim of this work was to optimize new extraction procedures. more environmentally sustainable and innovative, by comparison with classical methods. Finally, the polyphenolic composition and related antioxidant activity were also evaluated.

## 2. Materials and Methods

Ethanol, acetonitrile, bidistilled water, and *n*-hexane (HPLC-grade) were obtained from Merck Science Life s.r.l. (Milan, Italy). All solvents and chemical standards used in this paper were analytical grade products and were used without any further purification and were purchased from Merck Science Life s.r.l. (Milan, Italy). 

### 2.1. Samples

The pomegranate dried seeds (Granata cv) were kindly gifted by the biological company Giovomel (Avellino, Italia). The received seeds were, then, finely homogenized by a domestic mixer (Bimby Thermomix TM31, Vorwerk, Wuppertal, Germany). 

### 2.2. Soxhlet Extraction with n-Hexane

About 20 g of finely homogenized pomegranate seeds were extracted by Soxhlet apparatus [16] (500 mL capacity) with 500 mL of *n*-hexane for 4 h, at 79 °C. The obtained extraction solution was then evaporated by rotavapor (Soxhlet oil, **1**) and stored at 4 °C until further analyses were performed. 

### 2.3. CO_2_-EtOH Supercritical Extraction

According to Liu et al. (2009) [23] with substantial modifications, CO_2_-supercritical extraction was carried out with a supercritical extraction system (Jasco Europe s.r.l., Cremella, Italy). The system was characterized by a pump scCO_2_ Jasco-PU-4347 Unit, through a stainless-steel vessel, equipped with 10 µ porous stainless steel 316 L sintered filter disks. The vessel (10 mL capacity) was filled with the homogenized pomegranate seed (about 5.0 g) and heated using a Jasco CO-4065 oven (80 °C). Then, the CO_2_-BP-4390 (back pressure regulator) was set to the target pressure. After a pre-equilibrating time of 30 min, the supercritical extraction was performed in CO_2_/EtOH 90/10 *v/v* at 40 MPa, 5 mL/min flow rate, for 2 h. 

The obtained extraction solution was then evaporated by rotavapor (scCO_2_ oil, **2**) and stored at 4 °C until further analyses were performed.

### 2.4. Atmospheric Pressure Matrix-Assisted Laser Desorption/Ionization Mass Spectrometric (AP-MALDI-MS) Analysis

The Soxhlet oil, **1** and scCO_2_ oil, **2** were dissolved in *n*-hexane to a 0.5 mg/mL concentration and added to 25 µL of a sodium chloride solution (25 mg/mL). MALDI matrix solution was prepared by dissolving 20 mg of 2,5-DHB (2,5-dihydroxybenzoic acid) in 1 mL of H_2_O/CH_3_OH (1:9, *v/v*). For mass spectrometric analysis, 1 µL of the sample and 1 µL of the matrix solutions were stepwise spotted on the MALDI plate and dried at ambient temperature. Mass spectra were recorded in the 500–1000 *m/z* range using an AP-MALDI UHR ion source (MassTech, Inc., Columbia, MD, USA) coupled to an LTQ XL linear ion trap mass spectrometer (Thermo-Fisher Scientific, USA) with a resolution of 1500 FWHM due to the low intensity of the ionic signals. The source was equipped with an Nd:YAG laser with output at 355 nm operating with an energy of 4.6% to the full power and a repetition rate of 5000 Hz. Other source parameters were adjusted in the positive ion mode to optimize the ionic transmission. The capillary temperature was typically fixed at 300 °C. Xcalibur 2.0.6 software was used to acquire all the displayed mass spectra. 

The molecular weights expressed as mass-*to*-ratio (*m/z*) of all possible triacylglycerols (TAG) were calculated according to the equation below and assuming that these species are ionized by the addition of a sodium cation (Na^+^) [23,24]:*m/z* (TAG) = MW [glycerol − 3(OH)] + MW [(FA_1_ + FA_2_ + FA_3_) − 3(H)] + MW [Na^+^](1)

### 2.5. Transesterification Procedure

The oily residues (Soxhlet oil, **1** and scCO_2_ oil, **2**) were subjected to transesterification before being analyzed by GC/MS according to the following procedure.

A reactor equipped with a magnetic stirring bar was charged with 200.0 mg of oil and 0.4 mL of MeOH. Then, MeONa was added (48.2 mg, 0.9 mmol) and the mixture was stirred for 3 h at room temperature. After this time, the crude was diluted with AcOEt (50 mL) and washed with water (3 × 40 mL). The combined organic layers were dried over Na_2_SO_4_, filtered, and concentrated under vacuum obtaining pale-yellow transesterified oils (transesterified Soxhlet oil, **1a** and transesterified scCO_2_ oil, **2a**)

#### H and ^13^C-NMR Analysis

A value of 0.20 mL of oil samples (Soxhlet oil, **1** and scCO_2_ oil, **2**) and of transesterified samples (transesterified Soxhlet oil, **1a** and transesterified scCO_2_ oil, **2a**) were dissolved in 0.5 mL of CDCl_3_ (I.E% = 99.80%), transferred in an NMR tube, and analysed.

^1^H NMR (400.13 MHz), and ^13^C NMR (100.6 MHz) were recorded with a Bruker Avance 400 spectrometer (Milan, Italy), equipped with a Nanobay console and Cryoprobe Prodigy probe. The resulting ^1^H NMR and ^13^C NMR spectra were processed using Bruker TOPSPIN software. 

### 2.6. Polyphenols Separation by Solid-Phase Extraction

Pomegranate seed oil samples (Soxhlet oil, **1** and scCO_2_ oil, **2**) were subjected to solid-phase extraction (SPE) using a Discovery^®^ DSC-18 SPE Tube column (Merck Life Science, S.r.l., Milan, Italy), according to Cairone et al. (2021) [25]. The column was previously conditioned with *n*-hexane. About 4.0 g of oil were dissolved in 10 mL of *n*-hexane and loaded into the column. The column was eluted with 6 mL of *n*-hexane and 6 mL of acetonitrile which were discarded. A total of 6 mL of methanol was used for the extraction of the polyphenolic content. The obtained methanol fraction was dried under reduced pressure with a rotary evaporator, at a controlled temperature of 40 °C, weighed, and stored (SPE Soxhlet oil, **1b** and SPE scCO_2_ oil, **2b**) at 4 °C until HPLC-DAD analyses were performed.

### 2.7. HPLC-DAD Analysis

The polyphenol’s extracts (SPE Soxhlet oil, **1b** and SPE scCO_2_ oil, **2b**) were weighed, solubilized in methanol (0.5 mg/mL) and filtered with a Millex^®^—LG filter (Low Protein Binding Hydrophilic PTFE 0.20 µM Membrane) (Merck Science Life, S.r.l, Milan, Italy). The solutions were injected and analyzed with an HPLC-DAD (Perkin Elmer, Milan, Italy), equipped as reported in Cairone et al. (2022) [26]. The analyses were performed at 280 and 360 nm on a Luna RP-18 3µ column, in gradient from 100% solvent A (water at 5% formic acid) to 85% in 15 min, from 85 to 55% in 30 min and from 55 to 40% in 20 min, at a flow rate of 0.8 mL/min. 

Analytes were identified by comparing retention times and UV-Vis absorption spectra to those of authentic standards (Appendix A).

### 2.8. DPPH (2,2-diphenyl-1-picryl-hydrazyl) Assay 

According to Cairone et al. (2021) [25], with slight modifications, a propan-2-ol solution of DPPH, about 180 µM, was prepared. Then, 2.5 mL of this solution were added to 0.5 mL of propan-2-ol, stored in the darkness, and monitored by UV/VIS Lambda 25 spectrophotometer (Perkin Elmer Waltham, MA, USA), at the wavelength of 515 nm, until the absorbance value was stable. The value of the maximum absorption expressed by the radical was read after 30 min. Then, 40.0 mg/mL of oil solution (**1** and **2**) was prepared, and 0.5 mL were added to the cuvette along with 2.5 mL of the above-described DPPH solution. The initial absorbance at 515 nm was determined and the reduction of DPPH absorbance after 30 min. was evaluated. A calibration curve with gallic acid was constructed (y = −0.276 ln(x) + 0.6143) and the exerted antioxidant capacity was expressed as gallic acid equivalents.

### 2.9. Statistical Analysis

Each assay was replicated at least three times. Data are expressed as mean ± SD. Statistical significance (ANOVA) was determined using the XLStat software (New York, NY, USA).

## 3. Results

### 3.1. Extraction Procedures 

In this study, two industrial extraction methods were compared, and their relative yields, advantages, and disadvantages were evaluated. Initially, a classical Soxhlet approach was used, based on *n*-hexane (operative temperature of 69 °C), tested extraction times of 2–6 h, and chose 4 h as the best compromise between time/energy consumption and obtained yields (of about 15%). Our results agree with what is reported in the literature [15,16,17,18,19], that the problems related to the Soxhlet procedure remain, mainly due to the long extraction times, the required energy, and the use of polluting organic solvents. Additionally, a supercritical CO_2_ extraction method was studied. After the optimization of extraction conditions performed in presence of a 10% of co-solvent ethanol, it was possible to obtain an appreciable yield (about 9% vs. 15% gained in Soxhlet apparatus), in a shorter time (2 h vs. 4 h) using not toxic and recyclable CO_2_ as an extracting substance. Furthermore, differences in triglycerides composition and better anti-radical activity were observed as furtherly shown by the NMR analyses, and the DPPH assay.

### 3.2. AP-MALDI-MS Analysis

Appendix A displays the calculated monoisotopic masses and molecular weights of possible sodium adducts of triacylglycerol (TAG) combinations. As reported by Kaufman et al. [23], the 16:0, 18:0, 18:1, and 18:2 fatty acids are consistent with palmitic acid (**P**), stearic acid (**S**), oleic acid (**O**), and linoleic acid (**L**), respectively, whereas the 18:3 fatty acid can be associated to linolenic acid (**Ln**) or to one of its positional and geometric isomers. Among these, punicic acid (Pu; 9-cis,11-trans,13-cis) is known to be the most common 18:3 fatty acid in pomegranate oil (Pu = 83%) [24]. Therefore, with the acronym Ln we refer to the complete set of possible 18:3 fatty acids. 

The AP-MALDI mass spectrum of the pomegranate oil extract (Soxhlet oil, **1**) was reported in Appendix A. No significant differences were shown by the AP-MALDI analysis of the two differently obtained extracts. Only very weak signals of DAGs were observed in the 590–620 *m/z* range, whereas ionic species corresponding to TAGs can be highlighted in the 830–915 *m/z* region. The near absence of DAG species in this pomegranate oil extract is an exception to the results reported in the literature [26] according to which, DAG ionic signals are more represented than TAG peaks. Evidently, neither the Soxhlet nor the supercritical extraction conditions we applied, caused hydrolysis of the triglyceride component.

A close inspection of the TAG region of the mass spectrum (Figure 1) reveals the occurrence of the following TAG distribution: a poorly intense ionic series corresponding to [(16:0); (18:3); (18:3)] and [(16:0); (18:3); (18:2)] species in the 870–880 *m/z* range (*m/z* = 873.7 and 875.7, respectively), four major ionic peaks spread in the 890–910 *m/z* region (*m/z* = 895.7, 897.7, 899.7, and 901.7), and a minor presence of [(18:1); (18:0); (18:0)] and [(18:0); (18:0); (18:0)] TAGs (*m/z* = 911.8 and 913.8) in the 910–920 *m/z* range. For clarity, theoretical mass spectra related to the found TAGs are reported in Appendix A. 

Considering the prevalent fatty acids found in the pomegranate oil [27], the ionic peaks identified in the mass spectrum may correspond to PLnLn (*m/z* = 873.7), PLnL (*m/z* = 875.7), LnLnLn (*m/z* = 895.7), LnLnL (*m/z* = 897.5), OSS (*m/z* = 911.8), and SSS (*m/z* = 913.8). The ions at *m/z* 899.7 and 901.7 can be associated with different combinations of fatty acid residues (Appendix A). In the case of the species at *m/z* 899.7 two possible compositions are reported: [(18:3); (18:2); (18:2)] or [(18:3); (18:3); (18:1)], whereas for the ion at *m/z* 901.7 even three combinations are accessible: [(18:2); (18:2); (18:2)], [(18:3); (18:3); (18:0)], and [(18:3); (18:2); (18:1)]. Since the Ln: (18:3) are the most widespread fatty acids represented in the pomegranate oil, we may assume that the preferential combination is LnLnO for the ion at *m/z* 899.7 and LnLnS for the ion at *m/z* 901.7.

In addition to this, the mass spectrometric approach employed in this study does not allow us to discriminate between the linolenic acid isomers, that we have generically indicated as Ln, nor to assign the TAG stereospecific structure, meaning the sn position (1, 2, or 3) of each fatty acid on the glycerol backbone. The latter limitation does not affect the attribution of the structure of the TAGs characterized by a homogeneous composition, such as LnLnLn and SSS species, whereas in all other cases, only speculative considerations can be made, according to the literature [27]. Saturated fatty acids, in fact, are typically situated in primary positions, named sn1 and sn3, whereas mono- and polyunsaturated fatty acids are predominantly localized in the sn2 position.

### 3.3. NMR Analysis

In addition to AP-MALDI spectrometric analysis, ^1^H and ^13^C-NMR (inverse gated) spectrometric analyses were also conducted on the two obtained oils (Soxhlet oil, **1** and scCO_2_ oil, **2**) in order to define the qualitative fatty acids composition and to evaluate the relative differences between the two samples. In Appendix A the corresponding ^1^H and ^13^C-NMR spectra are reported. These show the typical triacylglycerol profile of vegetable oil [28] but, due to the signal’s overlapping, do not allow us to establish the difference in fatty acid composition between the two samples. 

Therefore, a transesterification step to produce methyl esters of fatty acids (FAMEs) was performed obtaining two samples (transesterified Soxhlet oil, **1a** and transesterified scCO_2_ oil **2a**) whose ^1^H and ^13^C-NMR spectra are shown in Appendix A.

^13^C inverse gated experiment allows estimating the relative ratios of different fatty acids, especially in the double bond region between 126 and 135 ppm (Figure 2). In this case, spectra allowed us to confirm the differences in the composition of CLNA isomers of the two analysed samples. 

Bulleted lists look like this: the further characterization of the fatty acid methyl esters was performed by careful analysis of the relevant spectra with direct comparison in the literature [29,30,31,32] and by ppm superposition with the external standard of oleic acid and linoleic acid. The applied technique made it possible to unequivocally identify the double bond position of our CLNA isomers, as well as confirm the presence of linoleic and oleic acid. The attribution was also made possible by the study conducted by Cao et al. (2007) [31]. The study, in fact, presents results related to CLNA isomers characterized in detail using ^1^H NMR, ^13^C NMR, ^1^H-^1^H COSY, ^12^C-^1^H NMR, and GC/MS in combination. 

The expansion of the region between 125 and 136 ppm of the ^13^C-NMR spectra, allowed identifying the presence of the characteristic signals of punicic acid (*cis*-9, *trans*-11, *cis*-13), suggesting its presence as the main CLNA isomer of both samples. In addition, α-eleostearic acid, (*cis*-9, *trans*-11, *trans*-13) catalpic acid (*trans*-9, *trans*-11, *cis*-13), linoleic acid, and oleic acid were detected in transesterified Soxhlet oil, **1a** and in transesterified scCO_2_ oil, **2a.** Conversely, β-eleostearic acid (*trans*-9, *trans*-11, *trans*-13) was mainly identified in **1a**, being represented only in trace in **2a**. 

In particular, the ^13^C NMR of punicic acid showed the following six peaks in the olefinic region: δ_C_, C9-132.6, C10-128.8, C11-127.9, C12-127.8, C13-128.8, C14-132.5. A higher amount of punicic acid was observed in sample **2a** (76% of total CLNA) than in sample **1a** (54% of total CLNA). On the other hand, the remaining CLNAs were about two to three times more abundant in sample **1a** with respect to sample **2a**. Among them, the following were identified: α-eleostearic, (δ_C_, C9-131.7, C10-128.7, C11-132.8, C12-125.9, C13-130.6, C14-135.1) about 14% in sample **1a** and 7% in sample **2a**; catalpic acid (δ_C_, C9-131.9, C10-126.1, C11-130.7, C12-128.7, C13-132.7, C14-134.9), 13% in sample **1a** and 5% in sample **2a**; *β*-eleostearic (δ_C,_ C9-134.4, C10-130.9, C11-130.5, C12-130.4, C13-130.8, C14-134.2), 7% in sample **1a** and 2% in sample **2a**. Linoleic acid (δ_C_, C9-130.2, C10-129.97, C12-128.0, C13-127.9), 6% in sample **1a** and 5% in sample **2a** and oleic acid (δ_C_, C9-130.0, C10-129.7), 6% in sample **1a** and 5% in sample **2a** were also quantified. All the obtained results substantially agreed with data from the literature [33], with the only exception of the CLNA isomers distribution, very likely not related to the seed oil composition but to the adopted extraction method.

In fact, we could observe that the two extraction methods led to a substantial difference in the profile of CLNAs, probably due to punicic acid isomerization in *trans*-9 and/or *trans*-13-isomers, which may occur in the more drastic condition of the Soxhlet extraction.

### 3.4. Solid Phase Extraction, HPLC-DAD and DPPH Analysis

To evaluate the phytocomplex profile of pomegranate seed oil, polyphenols were also subjected to solid phase extraction (SPE) with C18 cartridges with the aim to isolate the polyphenolic fraction. The extraction yield in sample **1b** (SPE Soxhlet oil) afforded only to 0.2 %, whereas the yield was 1.9% in sample **2b** (SPE scCO_2_ oil)**.** Considering the oil yields obtained by supercritical CO_2_ extraction (scCO_2_ oil, **2**, 9%) and Soxhlet apparatus (Soxhlet oil, **1**, 15%), an absolute yield of about 1.8% of polyphenols was gained with the binary system CO_2_/EtOH 90:10, whereas only 0.3% was obtained with *n*-hexane. Obviously, the presence of ethanol strongly promotes the polyphenols co-extraction along with the lipophilic phase, so affording an extract more interesting in terms of anti-radical activity while performing extraction with supercritical fluids. The obtained SPE extracts (**1b** and **2b**) were further subjected to HPLC-DAD analysis for the polyphenol’s characterization. The related chromatograms are reported in Figure 3. 

Other than different contents, HPLC-DAD analyses performed at 280 and 360 nm, showed a different polyphenols composition, characterized by a different profile at 280 nm and by the presence of flavonols in sample **2b**, not yet represented in sample **1b**.

Some hydroxycinnamic acids and flavonoids could be determined. The profile at 280 nm allowed the identification and quantification, in both samples **1b** and **2b,** of coumaric acid (1), carvacrol (2), and thymol (3). Quercetin 3-D-galactoside (4) and other not identified flavonols were shown at 360 nm in the only sample **2b**. No literature is available on polyphenols isolated by pomegranate seeds’ triacylglycerol oil, but only few studies are related to polyphenols directly extracted from seeds. Among these, Ambigaipalan et al. (2017) [34] reported the presence of phenolic acids, flavonoids, tannins, and anthocyanins, (in particular gallic, hydroxybenzoic, vanillic and caffeic acid, catechins, quercetin derivatives, and ellagic acids, which were expressed as total phenolic content of about 1.38 mg GAE/g of seed). Similarly, He et al. (2011) [35] confirmed the presence of these molecules, reporting catechin, reaching values between 0.5 and 2.0 mg/g of seed extract, as the most abundant one. In Table 1, the data of quantitative HPLC analyses are reported.

Total polyphenols (not identified by external standards) represent the sum of all the peaks found in the range between 30 and 40 min at 280 nm, which were quantified as chlorogenic acid equivalents. The total amount of polyphenols is much higher in sample **2b** than in sample **1b** (about 100 vs. 43 µg chlorogenic acid/g crude oil). Moreover, coumaric acid contents were about two times higher in sample **1** (Soxhlet oil) than in sample **2** (scCO_2_ oil) (7.1 vs. 3.6 µg/g crude oil). Flavonoids, identified at 360 nm in the only sample **2b**, were quantified as the sum of the peaks found between 25 and 47 min at 360 nm and were expressed as quercetin-3-galactoside equivalents (16.4 µg/g crude oil). 

Terpenes phenols such as carvacrol and thymol, identified using pure reference standards, reached very interesting values, especially in sample **2b** (thymol about 470 µg/g and carvacrol 137 µg/g of crude oil) largely afforded to the best-represented class of compounds (as shown by Appendix A). Since there are no reports in the literature about the presence of phenolic terpenes in pomegranate seed oil, this research opens up new ground for further study and investigation. The higher content of terpenes, phenols, and phenolic acids, as well as the presence of flavonoids, found only in sample **2b**, demonstrate the best performance of supercritical CO_2_ extraction, assisted by ethanol as co-solvent, also in consideration of the mild operating conditions and the reduced extraction times.

### 3.5. Antioxidant Activity

The DPPH assay was performed directly on Soxhlet oil, **1** and **scCO_2_, 2** to evaluate their antiradical activity. The results, expressed as mg gallic acid equivalents/g oil (GAE), are shown in Figure 4, as mean values of three different experiments.

In total agreement with the results shown by the HPLC-DAD analysis, performed on samples **1b** (SPE Soxhlet oil) and **2b** (SPE scCO_2_ oil), sample **2** (scCO_2_ oil), presented an exceptionally high antiradical activity (about five-fold, with respect to the adopted reference of extra virgin olive oils, EVOOs, [25]), as well as sample **1** (Soxhlet oil) presented a DPPH value only slightly higher respect to EVOOs (1,2:1).

These results are confirmed by other results found in the literature. In particular, in a study by Abbasi et al. (2008) [36], a higher phenolic content was observed in pomegranate seed oil extracted by supercritical CO_2_ if compared to Soxhlet, microwave, and/or ultrasonic extractions. The exceptionally high activity here we found, is surely due to the adopted extraction conditions, based on the binary system CO_2_/ethanol, which leads to a greater selection for polyphenols and corresponds to a more efficient anti-radical extract [37]. 

## 4. Conclusions

Pomegranate seeds, usually considered waste, were valorized for their triacylglycerols and polyphenols content. Two differently obtained extracts showed a different composition in CLNA isomers, in polyphenol content and composition as well as in anti-radical exerted activity. Punicic acid, actually under evaluation for its effect on eicosanoid production and suppression of inflammatory response, reached a prevalence of 76% in the extract obtained by supercritical fluids (CO_2_ and ethanol), with respect to a content of 54% in Soxhlet (*n*-hexane). Polyphenols were also significantly more abundant in the extract obtained with supercritical fluids. If the predominance of punicic acid was due to its better preservation under milder extraction conditions, the higher extraction of polyphenols was due to the polarity of the co-solvent ethanol. On the whole, the oily phase obtained by supercritical extraction of pomegranate seeds showed a better composition, corresponding to a potent anti-radical activity, with about five-fold respect to EVOOs used as references. 

Despite the higher capital and investment costs, which hinder the commercialization of CO_2_-extracted vegetable oils, these results highlight the possibility to obtain highly valuable functional foods, not only avoiding waste production but generating healthy supplements. The excellent results obtained by using a mixture of CO_2_ and 10% ethanol on this matrix, open new scenarios for the use of other polar solvents. Different ratios with carbon dioxide and different operative conditions will deserve to be tested, with the aim to valorise other food waste not yet explored.

## Figures and Tables

**Figure 1 foods-12-01592-f001:**
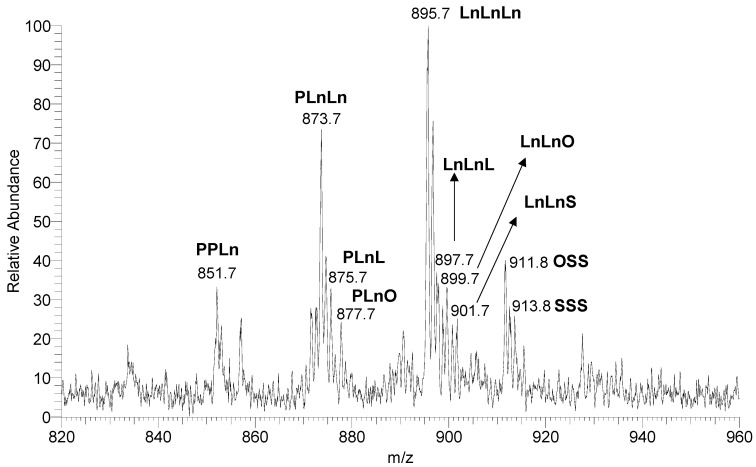
Amplification of the TAG region in the 820-960 mass range of the AP-MALDI mass spectrum.

**Figure 2 foods-12-01592-f002:**
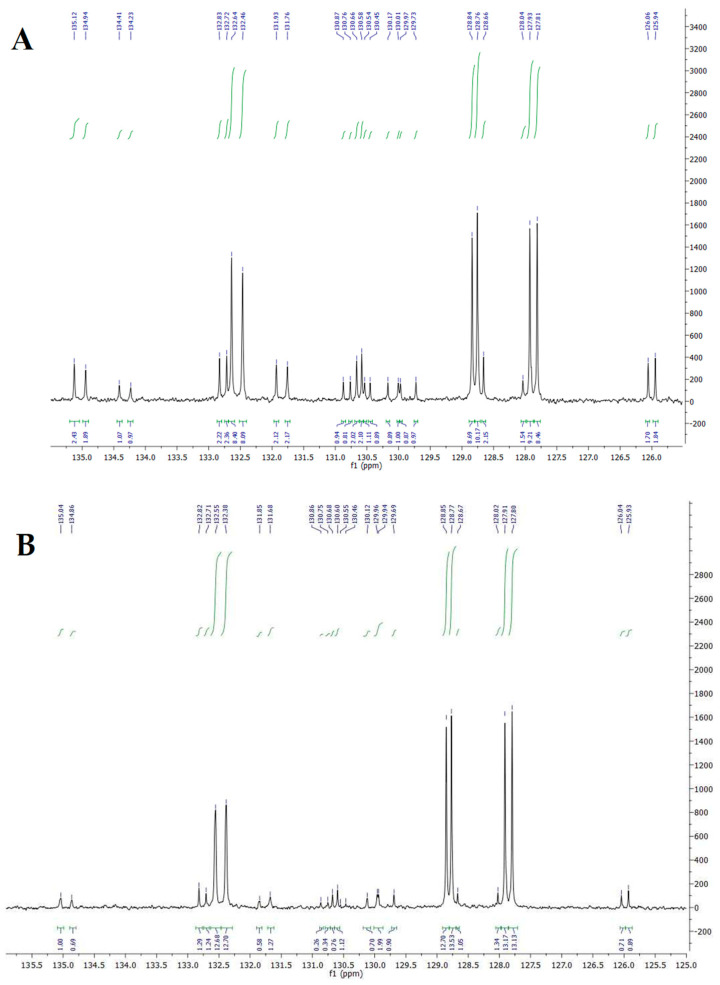
^13^C NMR diagnostic olefinic region related to transesterified Soxhlet oil, **1a**, (**A**), transesterified scCO_2_ oil, **2a** (**B**).

**Figure 3 foods-12-01592-f003:**
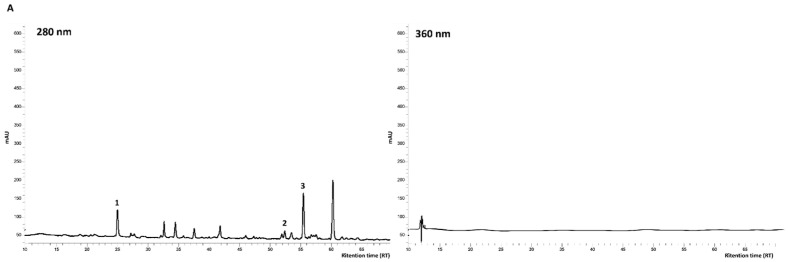
Example chromatograms of polyphenols fraction. (**A**): Sample **1b** (SPE Soxhlet oil), polyphenolic profile at 280 and 360 nm; (**B**): Sample **2b** (SPE CO_2_ oil) polyphenolic profile at 280 and 360 nm. Identified peaks: 1. Coumaric acid; 2. Carvacrol; 3. Thymol; 4. Quercetin-3-D-galactoside.

**Figure 4 foods-12-01592-f004:**
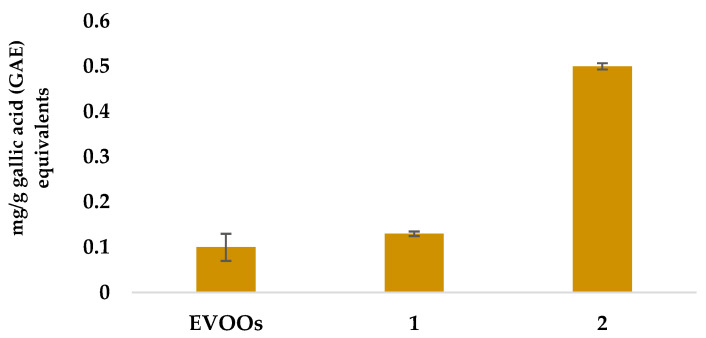
Antiradical activity of Soxhlet oil, **1** and scCO_2_ oil, **2**, compared to a mediated result obtained from 45 samples of EVOOs. Results are reported as GAE mg/g oil.

**Table 1 foods-12-01592-t001:** HPLC-DAD quantitative analysis of the SPE Soxhlet oil, **1b**, and scCO_2_ oil, **2b**. Results are expressed in µg/g of crude oil.

Compound	1b	2b
Coumaric acid	7.14 ± 0.22	3.61 ± 0.02
Carvacrol	20.50 ± 0.47	136.77 ± 1.33
Timol	108.63 ± 5.88	471.52 ± 2.60
Total polyphenols ^1^	43.49 ± 1.52	99.57 ± 2.98
Flavonols ^2^	- ^3^	16.37 ± 0.33

^1^ Expressed as chlorogenic acid equivalents; ^2^ expressed as quercetin-3-galactoside equivalents ^3^ not detected.

## Data Availability

The data used to support the findings of this study can be made available by the corresponding author upon request.

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
