# Peer review of "In-Depth Chemical Characterization of Punica granatum L. Seed Oil"

_foods, 2023, doi:10.3390/foods12081592_

Round 1

Reviewer 1 Report

The authors thoroughly characterized chemically the seed oil of Punica granatum L. The paper presents interesting results, which is of interest for a growing range of industrial and food processing, the literature review is very comprehensive it is reasonably well written and of interest of the journal. However it needs minor corrections, as follows:

1-      Why you used 2 styles of bibliographic references (numbering and at the same time by names of authors and years);

2-      You have not presented the originality of your work. Try to reformulate the end of your introduction and specify the originality and the objective of your work;

3-      Why you presented only the extraction method (Soxhlet), are there other methods of extraction and why you did not mention;

4-      Similarly to figure 3, the abscissa and ordinate values are not visible;

5-      Standardize the references, as an example the ref [1],  you have ignored number of pages.

Reviewer 2 Report

The topic is very interesting, but it has been covered many times.
In the introduction, the health aspect is detailed, with a minor presentation of the bioactive constituents of this particular oil. The authors wander from topic to topic. The focus should be on the constituents of the oil extracted from the seeds.
It would be more interesting if the oil obtained by cold pressing was also compared.
Supplementary material (tables) is not shown in the paper, so it is impossible to comment on a significant part of the work.
The methods are novel, but in commenting on the results, some important references on the subject of this oil are omitted, such as:

Celia Bañares, Assamae Chabni, Blanca Pardo de Donlebún, Guillermo Reglero, Carlos F. Torres, Chemical characterization of pomegranate and alfalfa seed oils obtained by a two-step sequential extraction procedure of expeller and supercritical CO2 technologies, Journal of Food Composition and Analysis, Volume 115, 2023, 105040, ISSN 0889-1575, https://doi.org/10.1016/j.jfca.2022.105040.

Andrea Natolino, Carla Da Porto, Supercritical carbon dioxide extraction of pomegranate (Punica granatum L.) seed oil: Kinetic modelling and solubility evaluation, The Journal of Supercritical Fluids, Volume 151, 2019, Pages 30-39, ISSN 0896-8446, https://doi.org/10.1016/j.supflu.2019.05.002.

Ahangari, B., Sargolzaei, J. Extraction of pomegranate seed oil using subcritical propane and supercritical carbon dioxide. Theor Found Chem Eng 46, 258–265 (2012). https://doi.org/10.1134/S0040579512030013

Zinar Pinar Gumus, Zeliha Ustun Argon, Veysel Umut Celenk, Chapter 53 - Cold pressed pomegranate (Punica granatum) seed oil, Editor(s): Mohamed Fawzy Ramadan, Cold Pressed Oils, Academic Press, 2020, Pages 597-609, ISBN 9780128181881, https://doi.org/10.1016/B978-0-12-818188-1.00053-0.

ZieliÅ„ska A, Wójcicki K, Klensporf-Pawlik D, Marzec M, Lucarini M, Durazzo A, Fonseca J, Santini A, Nowak I, Souto EB. Cold-Pressed Pomegranate Seed Oil: Study of Punicic Acid Properties by Coupling of GC/FID and FTIR. Molecules. 2022 Sep 9;27(18):5863. doi: 10.3390/molecules27185863. PMID: 36144599; PMCID: PMC9501365.

Boroushaki, M.T., & Mollazadeh, H. (2016). Pomegranate seed oil: A comprehensive review on its therapeutic effects.

Other comments are given in the manuscript.

Reviewer 3 Report

Dear Authors,

I had the pleasure of reviewing your article. The idea of performing such research is very interesting because of the small accessibility of information about oil extras in CO2. The manuscript is written well but requires a few corrections.

1. Please remove citation in brackets.

2. Line 73 why do you use the designation for CLnA?

3. Are used methods yours or other researchers? You need to provide a citation.

4. Line 179 Please give the unie name for 30'.

5. Statistic analysis: have you done any statistical analysis? Homologues grups, Anova?

6. Line 315 use italic for β.

7. Oil markings are not very intuitive; I suggest replacing them with those that will suggest a kind of oil or conducted extraction.

8. Table 1 is not in line with the requirements.

9. Table 1 you analyse polyphenols or phenolic compounds?

10. Paragraph 3.5 change the name of the DPPH analysis to antioxidant activity.

11. Fig 5 - what unit should antioxidant activity have? mg of galicic acid/g of oil?

Round 2

Reviewer 2 Report

Since the authors have accepted all suggestions and improved the manuscript, I propose to publish it.